# Comparison of the Performance of the Leap Motion Controller^TM^ with a Standard Marker-Based Motion Capture System

**DOI:** 10.3390/s21051750

**Published:** 2021-03-03

**Authors:** Amartya Ganguly, Gabriel Rashidi, Katja Mombaur

**Affiliations:** 1Optimization, Robotics and Biomechanics, Institute of Computer Engineering, Heidelberg University, 69120 Heidelberg, Germany; g.rashidi@stud.uni-heidelberg.de; 2Canada Excellence Chair in Human-Centred Robotics and Machine Intelligence, University of Waterloo, Waterloo, ON N2L 3G1, Canada; katja.mombaur@uwaterloo.ca

**Keywords:** Leap Motion Controller™, motion capture, finger kinematics, Bland–Altman analysis

## Abstract

Over the last few years, the Leap Motion Controller™ (LMC) has been increasingly used in clinical environments to track hand, wrist and forearm positions as an alternative to the gold-standard motion capture systems. Since the LMC is marker-less, portable, easy-to-use and low-cost, it is rapidly being adopted in healthcare services. This paper demonstrates the comparison of finger kinematic data between the LMC and a gold-standard marker-based motion capture system, Qualisys Track Manager (QTM). Both systems were time synchronised, and the participants performed abduction/adduction of the thumb and flexion/extension movements of all fingers. The LMC and QTM were compared in both static measuring finger segment lengths and dynamic flexion movements of all fingers. A Bland–Altman plot was used to demonstrate the performance of the LMC versus QTM with Pearson’s correlation (*r*) to demonstrate trends in the data. Only the proximal interphalangeal joint (PIP) joint of the middle and ring finger during flexion/extension demonstrated acceptable agreement (*r* = 0.9062; *r* = 0.8978), but with a high mean bias. In conclusion, the study shows that currently, the LMC is not suitable to replace gold-standard motion capture systems in clinical settings. Further studies should be conducted to validate the performance of the LMC as it is updated and upgraded.

## 1. Introduction

The Leap Motion Controller™ (LMC) device is a portable, low-cost marker-less motion capture system and has been progressively used to replace gold-standard marker-based motion capture systems, especially in clinical environments. The LMC is designed to track elbow, wrist and finger joint positions. It is used in a myriad of different scenarios, for example human-robot interaction [1,2,3], surgery training [4] and gesture recognition [5,6]. The LMC has also found application in aiding hand rehabilitation; for example, it has been used as a development and evaluation system for post-stroke patients [7] and used for video-game-based therapy in the upper extremities [8,9,10]. In another study [11], the LMC was used to investigate hand tremor, a common movement disorder, often found in patients with dystonia, Parkinson’s disease, and cerebellar ataxia. While the LMC has uses in various hand rehabilitation therapies, it is also used as a measurement device. The validation of such marker-less technology is therefore warranted because it has found application in such diverse areas. Over the years, a number of validation studies have been conducted, but with mixed results.

Initially, the LMC was validated for point or target-based implementations, as the device primarily was designed for gesture-based control. In this regard, endpoint tracking, via a reference pen, was used to validate the LMC with an accuracy of 0.2 mm. The study argued that an accuracy of <2.5 mm is possible with an average of 1.2 ± 0.7 mm [12]. One study performed a Fitts’ tapping task and reported high error rates, where the percentage of out-of-target selections, for the LMC, was 7.2% versus a standard mouse device with 2.3%. They also showed that the movement time was significantly longer with the LMC compared with the mouse movements (945 versus 565 ms). However, the study did not synchronise or simultaneously record the trials with the mouse and the LMC [13].

There were only a few studies that attempted to validate the LMC against standard marker-based motion capture systems. In one of the first studies, they restricted themselves to the wrist and the forearm movements [14]. The results showed that the LMC was unreliable for forearm pronation/supination movements. However, for wrist flexion/extension and deviation, it had acceptable performance, suitable within a 10% threshold region, as reported in studies with electronic goniometers [14,15,16]. In a separate study, the temporal characteristics were compared, where the LMC was deemed reliable for movement time and deceleration interval parameters. However, spatial measurement parameters such as endpoint accuracy, precision and maximum grip aperture were unreliable [17]. In another study, LMC when compared against the Optotrak motion capture system had a high correlation, 0.995 on the horizontal axis and 0.945 on the vertical axis, with the Optotrak. The finger accuracy of the LMC was 17.3 ± 9.56 mm versus the Optotrak [18]. None of these motion capture investigations focused on finger kinematics in any way.

To our knowledge, there have been only two validation studies in which the LMC was compared against goniometer measurements for finger kinematics. One study compared finger abduction and flexion of the Metacarpophalangeal (MCP) joints using both LMC and manual goniometers [19]. It was found that abduction angles between the index and middle finger were in good agreement; however, ring and the little finger abduction angles were less reliable. In the case of MCP flexion angles, only the middle finger showed acceptable agreement [19]. A more thorough study, with a larger sample size, was conducted, where it was demonstrated that the measurements between the LMC and the goniometers were not agreeable [20].

It is evident that the LMC is being used in a variety of applications; however, there is a lack of validation and replication studies with regards to clinical acceptance of this low-cost marker-less device. Specifically, there is a gap in the evaluation of the performance of the LMC with respect to finger kinematics, as well as a lack of synchronised validation of the LMC with a gold-standard motion capture system. To the best of our knowledge, this is the first validation study to synchronise a gold-standard marker-based motion capture system with the LMC with respect to finger kinematic assessment. In this paper, we perform both the static measurement of finger segment lengths and dynamic conditions measuring the joint angles, for all five fingers for the flexion and extension movements. The experiment was designed in a manner that can be reproduced in a clinical setting.

## 2. Materials and Methods

The subjects who volunteered for the this study were recruited from the university campus. They were deemed healthy and free from injury or neurological disorders. All subjects agreed and signed a written informed consent prior to any experiments. The experimental protocol and practices were performed with the complete approval of the Faculty of Behavioural and Cultural Studies, University of Heidelberg Ethics Committee.

### 2.1. Subject Information

Ten right-handed subjects (6 male and 4 female) of a mean age of 26.8 ± 4.56 years were recruited for this study. The experimental protocol was completed without incident, and all subjects successfully performed the assigned tasks.

### 2.2. Marker-Based Motion Capture Technology

Ten Oqus 500 motion capture cameras (Qualisys Track Manager (QTM); Qualisys, Gothenburg, Sweden), placed at appropriate locations, were used at a frame rate of 150 Hz. Each subject wore 24 retro-reflective markers of 4mm in size on their right hand (dominant hand), as shown in Figure 1a. These markers were carefully placed on the bony landmarks of the hand. The use of this marker set was previously validated in [21] with a mean repeatable accuracy of 5.1${}^{\circ }$. The motion capture system tracks these markers in the given XYZ space. The standard calibration practice, as detailed in the instruction manual, was used. A calibration was acceptable if the average residuals were 0.6mm, and data collection commenced only after a successful calibration.

### 2.3. Leap Motion Controller™

The LMC (Ultraleap, Bristol, United Kingdom) renders a skeletal model of the hand where the joint centres are similar to the markers placed for the QTM system, as depicted in Figure 1b. It is a marker-less, motion capture device, as shown in Figure 1c, which incorporates infrared and stereo vision technologies. The device weighs 32 g and is rectangular in shape with dimensions of 30 mm × 11.30 mm × 80 mm. It has three light emitting diodes (LEDs) spaced in between the two stereo cameras. The cameras are spaced 40 mm apart with a resolution of 640 × 240 pixels. It typically operates at 120 Hz. The data are streamed via a USB2.0/3.0 hybrid cable. The device works on Windows, Macintosh and Linux platforms. The controller tracks hands within a 3D interaction area that extends up to 60–80 cm from the device in a typical field of view of 140 × 120 degrees. The LMC generally is pre-calibrated; however, it was calibrated again with a calibration score of 95. A higher calibration score yields better stable measurements [22]. The LMC does not provide a depth scene of any kind or emit any form of structured light. A proprietary algorithm performs the calculation to obtain the position of the joints/objects from stereo-vision images. The LMC skeletal model is generated from a proprietary algorithm, which cannot be changed by a third party. This algorithm estimates the joint centre positions of the hand.

### 2.4. Data Collection

The two data acquisition systems were time synchronised because each system streams data with a different sampling frequency. So far, studies [12,14,20] have recorded data simultaneously with other measuring systems or have measured data independently. The LMC module has a constantly evolving Software Developing Kit (SDK) which currently provides support for Java, C and Python. Python was used as the programming language of choice. The data acquisition code along with the software synchronization of the LMC and QTM can be found via the link given the Data Availability Statement. The associated animations also capture the exact nature of the flexion movements.

All participants performed a static trial and a series of flexion/extension tasks. Instructions were prompted on the screen, and all tasks were randomised. Two trials per subject were conducted. In each trial, the subjects performed flexion/extension movements three times. Overall, there were 10 successful static trials, and out of 120 dynamic trials, 103 were successful. The unsuccessful trials were attributed to LMC failure to synchronize with the QTM system or the system failure of either the LMC or QTM.

### 2.5. Data Processing

The LMC is known for streaming data with a variable sampling frequency [14,17]. In order to overcome this problem, a software program was written in Python to synchronise the data acquisition of QTM and the LMC. However, upon initial review of the data, it was revealed that the LMC data streaming was variable, but the time taken to complete the trials closely matched the QTM system. The temporal alignment of the LMC will not influence the real-world angular accuracy, and analyses of the temporal precision are not the objective of this study.

Since frame per second data vary, a new time vector was calculated. For *i* samples, the time would be $1/f_{i}$, and $f_{i}$ was estimated as the difference between samples i−1 and *i*. The new time vector would be the aggregation of the time intervals between 0 and *i* samples. The LMC data are interpolated over this new time vector, and the data are re-sampled at 150 Hz, matching the QTM data acquisition sampling rate.

### 2.6. Data Analysis

Figure 2 shows the geometrical diagram of a finger during a flexion/extension movement. The joint angles from the LMC and QTM were calculated from their recorded joint centre positions using the following set of equations:(1)$$\theta_{M}=\cos^{-1}\left(\frac{\vec{WM}\cdot \vec{MP}}{∥\vec{WM}∥\cdot ∥\vec{MP}∥}\right)$$
(2)$$\theta_{P}=\cos^{-1}\left(\frac{\vec{MP}\cdot \vec{PD}}{∥\vec{MP}∥\cdot ∥\vec{PD}∥}\right)$$
(3)$$\theta_{D}=\cos^{-1}\left(\frac{\vec{PD}\cdot \vec{DT}}{∥\vec{PD}∥\cdot ∥\vec{DT}∥}\right)$$
where the operator ∥.∥ is the magnitude of the designated vector.

Calculation of the segment length was performed by taking the Euclidean distance of the joint centres of each phalange.

The Bland-Altman (BA) plot is a good method to determine the effectiveness of a new method/measurement to substitute an established or gold-standard method/measurement. In this paper, the LMC, a low-cost, portable hand tracking module, is evaluated against a more expensive, but gold-standard motion capture system, QTM. There have been other studies that have used a difference or the BA plot to show their results pertaining to the LMC versus the motion capture system [14,19,20].

## 3. Results

### Initial Analysis

The study was divided into static and dynamic analysis. In the static case, the segment lengths of each subject were calculated and compared. The calculation method is described in Section 2.5. Note that the LMC estimates the joint centre position through a skeletal model that cannot be changed by a third party as access is not granted to make any changes to the algorithm. The QTM on the other hand tracks the markers placed on the bony landmarks of the hand. Therefore, the LMC measured joint positions are based on the accuracy of the inherent skeletal model of the device and not the camera of the system. This is in contrast to the QTM system, where each marker is visualised by at least three cameras. This makes the tracking in the captured volume more accurate relative to the laboratory origin.

In the dynamic case, the range of motion (ROM) of every movement per finger was analysed. A standard BA plot was used to compare the LMC measurements against the gold-standard QTM measurements. A trend of the results for each joint is shown along with the correlation coefficient. The bias, as well as the 95% confidence interval (CI) were also calculated to render a complete observation of the results.

Figure 3 shows the calculated segment lengths of ten subjects. The LMC data provide metacarpal base and height positions; however, for QTM measurements, the metacarpals were not considered apart from the thumb. The thumb has only three bones, metacarpal, proximal and distal phalanges, whereas the rest of the fingers have four. It is observed that each measuring system on its own showed little difference. However, the overall mean and standard deviation of the study are given in Table 1. The table shows that for the thumb, the LMC recordings were higher than QTM for all segments (distal, proximal and metacarpal). However, the QTM recordings were higher for the proximal and intermediate phalanges of the index, middle, ring and pinky fingers, respectively.

Figure 4 shows the flexion angles’ trajectories computed for the LMC and QTM systems for the MCP, Proximal interphalangeal joint (PIP) and Distal interphalangeal joint (DIP) joints of the index finger, respectively. For the MCP joint, in the case of QTM, the maximum angle measured was 78.01${}^{\circ }$ and the minimum 9.62${}^{\circ }$, whereas, in the case of the LMC, 80.85${}^{\circ }$ was the maximum and 9.69${}^{\circ }$ the minimum flexion angle measured. For the PIP joint, QTM measured a maximum flexion angle of 73.80${}^{\circ }$ and a minimum angle of 2.33${}^{\circ }$, whereas the LMC measured 83.40${}^{\circ }$ as the maximum and 2.62${}^{\circ }$ as the minimum flexion angle. For the DIP joint, the QTM measured a maximum flexion angle of 17.88${}^{\circ }$ and a minimum of 1.63${}^{\circ }$, whereas the LMC measured 70.12${}^{\circ }$ as the maximum and 1.68${}^{\circ }$ as the minimum flexion angle. While for the MCP and PIP joints, the amplitudes of the trajectories from the two systems were similar, the LMC measured angle for the DIP joint was grossly over-estimated. The two systems where time synchronised by starting and ending the trials at the same time programmatically, the time difference between the two systems for this trial was 1.13 s.

Table 2 compares the range of motion calculated from the LMC and QTM measuring systems. It is seen that the difference in the calculated ROM for the interphalangeal joint (IP) joint and MCP during thumb flexion is high, 28.29${}^{\circ }$ and 19.31${}^{\circ }$, respectively. The difference in the ROM of the PIP joint during flexion of the middle finger is the highest at 61.51${}^{\circ }$, which shows that the LMC grossly underestimated the flexion angles during this movement. The table shows that for thumb abduction and index finger flexion, the difference between the LMC and QTM was relatively low. However, in other cases, the ROM of the DIP joint for the ring finger and the PIP joint of the ring and pinky, the differences were relatively high.

The results shown in Figure 4 show the following:(1)The time recorded for the trials by the two systems varied. This shows the need to compare dynamic trials using techniques that will handle such time dependent data. In this paper, we used time normalisation to compare flexion trials between the LMC and QTM.(2)Table 2 shows that the ROM calculation for each joint during the flexion trials was quite varied, and this information is not enough to conclusively say that LMC performed better or worse than the QTM system. Therefore, subsequent results compared the LMC and QTM with the help of BA plots.

Figure 5A shows a BA plot for CMC joint with a negative bias of 28.14${}^{\circ }$ with r=0.1385, P=0.6088. This shows that there is a weak correlation and an increase in the difference with the magnitude of the measurement. Figure 5B shows a positive bias of 7.72${}^{\circ }$ with r=−0.0777, P=0.7747 of the MCP joint. Lastly, Figure 5C shows a negative bias of 4.94${}^{\circ }$ with r=0.1094, P=0.6865 of the IP joint. During the thumb abduction movement, none of the three thumb joints demonstrated a high correlation significance between the QTM and LMC measuring systems. It seems that for thumb abduction/adduction movement, there is no agreement for the joint angles of the MCP and IP joints.

Figure 6A shows a BA plot for the CMC joint with a negative bias of 31.31${}^{\circ }$ with r=0.0084, P=0.9735. Figure 6B shows a positive bias of 9.19${}^{\circ }$ with r=0.6381, P=0.0043 for the MCP joint. Lastly, Figure 6C shows a positive bias of 2.01${}^{\circ }$ with r=0.3219, P=0.1926 for the IP joint. During the thumb flexion movement, only the flexion of the MP joint demonstrated the highest correlation significance between the QTM and LMC measuring systems. However, the CMC joint showed the least correlation.

Figure 7A shows a BA plot for the MCP joint with a negative bias of 1.25${}^{\circ }$ with r=−0.3963, P=0.1152. Figure 7B shows a negative bias of 1.1${}^{\circ }$ with r=−0.1062, P=0.6894 for the PIP joint. Lastly, Figure 7C shows a positive bias of 0.1${}^{\circ }$ with r=−0.0384, P=0.8835 for the DIP joint. During the index finger flexion movement, none of the joints demonstrated a positive correlation between the two measurement systems.

Figure 8A shows a BA plot for the MCP joint with a positive bias of 10.67${}^{\circ }$ with r=0.0116, P=0.9646. Figure 8B shows a positive bias of 14.92${}^{\circ }$ with r=0.9062, P<0.001 fir the PIP joint. Lastly, Figure 8C shows a negative bias of 3.51${}^{\circ }$ with r=0.4352, P=0.0815 for the DIP joint. During the middle finger flexion movement, the PIP joint demonstrated the highest positive correlation between the two systems.

Figure 9A shows a BA plot for the MCP joint with a positive bias of 4.83${}^{\circ }$ with r=0.3679, P=0.1608. Figure 9B shows a positive bias of 18.76${}^{\circ }$ with r=0.8978, P<0.001 for the PIP joint. Lastly, Figure 9C shows a positive bias of 2.67${}^{\circ }$ with r=0.4137, P=0.1111 for the DIP joint. During the ring finger flexion movement, all joints showed positive correlations, as well as positive bias; however, the highest correlation was observed for the PIP joint.

Figure 10A shows a BA plot for the MCP joint with a positive bias of 13.64${}^{\circ }$ with r=0.2378, P=0.3749. Figure 10B shows a positive bias of 10.97${}^{\circ }$ with r=0.6676, P=0.0047 for the PIP joint. Lastly, Figure 10C shows a positive bias of 4.75${}^{\circ }$ with r=0.5736, P=0.0201 for the DIP joint. During the little finger flexion movement, all joints showed positive correlations, as well as positive bias; however, the maximum correlation was observed for the PIP joint.

## 4. Discussion

In earlier studies, regarding the accuracy of the LMC for static and dynamic tracking, a mannequin arm was used, where, for the static case, the standard deviation was less than 0.5 mm. However, with dynamic tracking, the LMC was inconsistent in performance, with the accuracy reduced by 5 mm [23]. The static results reported in this paper (Figure 3) are consistent with [23]. The time synchronisation revealed more stable data collection during the static case; however, there was still some variability for the dynamic trials.

There has not been any study so far that has validated the LMC against marker-based motion capture with respect to finger kinematics; however, there have been other measuring devices such as the goniometer [16,20]. The LMC was not in agreement with the goniometric readings for most of the joints with the exception of the wrist and the MCP of the index finger [20]. Overall, it was concluded that the LMC was not yet ready as a clinical measurement tool, and this conclusion matches with the overall findings of this paper.

### 4.1. Why Is a Trend Observed between Two Measurement Techniques?

In such comparative studies, it is possible to observe a clear trend where the difference increases with the increase in the magnitude of the parameter being measured, in this case the ROM. This consequently shows that the variances would be different, which was evident with the different biases reported for every joint with respect to their fingers. The results show that the standard deviation of the mean difference between the LMC and QTM is high, because the LMC measurement is lower than QTM for low values, as seen in the distal joints, and greater than QTM for high values, as seen in the MCP joints. This is the primary reason why a non-zero correlation between the difference and the mean was observed. However, the thumb MCP joint (Figure 5B), the thumb CMC joint (Figure 6A), the index finger DIP joint (Figure 7C) and the middle finger MCP joint (Figure 8A) showed a negligible correlation of little significance.

### 4.2. Repeatability

Repeatability is an important criterion in a study that involves method comparison, such as validation studies between the LMC and gold-standard motion capture technology. This is because the repeatability of two different methods of measurement has a probability to limit the amount of their agreement. If one of the methods has a poor repeatability, significant variation is expected in the same subject. To mitigate such a scenario in this study, the subjects were allowed to flex/extend three times to achieve the maximum functional ROM of their fingers. However, the LMC showed a considerable difference during these attempts, and there were instances when the trial failed because of synchronisation failure. Therefore, further trials with a larger participant cohort are necessary to take into account the repeatability criteria in order to conclusively validate the performance of the LMC with respect to standard motion capture technology.

### 4.3. Clinical Recommendation

In the static case, where the hand is poised over the LMC device with negligible movement, the LMC is able to measure the joint centre positions on each finger segment efficiently. The subsequent calculation of the segment length showed that the LMC is reliable with a low standard deviation, as shown in Figure 4. However, in the case of the dynamic condition, it was evident from the results that the LMC was not reliable enough to effectively track the movements of the fingers. The ring finger and the little finger showed weak correlations with QTM at all joint levels. Overall, the results for the thumb, index and middle finger were not promising either. Currently, the LMC cannot be recommended to monitor finger ROM in a clinical setting. However, from previous studies, it has been demonstrated that the LMC can be used to monitor wrist ROM, demonstrating acceptable accuracy within a 10% threshold, reported by goniometric measurements [14,15,16,20].

Before recommending the LMC, as an alternative to gold-standard motion capture or goniometers, further research is required to evaluate the performance of the device to effectively capture finger kinematic data in an unsupervised clinical environment.

### 4.4. Study Limitations

Our attempt to synchronise the start and end time of data acquisition was found to be suitable only for the static conditions, where the LMC performed well. However, for the dynamic conditions, we encountered various inherent problems that prevented a more reliable data capture. One of the main limiting factors was the variable frequency of the LMC during measurements as opposed to the constant frequency for QTM. The variable frequency was found to be mainly in dynamic trials because the total duration of the trial was longer than static trials. This affected the calculation of the flexion/extension and abduction angles for the finger joints. The LMC algorithm does not directly measure joint positions, but estimates them; however, it has been shown that the endpoint positions seem more reliable than joint centre positions [24]. It has been previously reported that the measurement of finger joint angles is susceptible to large variability [25]. Therefore, an acceptable clinical error needs to be established for joints of the fingers, as has been done for the wrist and forearm [15,16], and to determine an effective range.

The synchronisation program for the LMC and QTM was dependent on whether the two programs were run on two different machines. However, in this case, a loop-back interface was used to minimise the delay in the signal between the two data loggers. If the signal was weak or there was a momentary break, this would affect the data collection; hence, some of the trials were classed as unsuccessful because of synchronisation failure. The presence of retro-reflective markers on the subjects may have contributed to the unreliable LMC data even though the marker size was as small as possible to mitigate such a scenario. These markers are crucial for any standard motion capture system as they reflect infrared light. The LMC also emits infrared light; hence, this may cause disruption during the recordings.

In this study, we attempted to record the flexion/extension movements of our participants; however, the study protocol was restrictive. The participants were seated in a standard seat with an appropriate height and distance from a table. The LMC was fixed at an optimal reaching distance for each participant. Although the tasks were randomised, each participant followed a cue on a screen; however, the speed of the each movement was not fixed, and the height of the hand from the LMC varied within each trial. These variations were incorporated based on recommendations by a previous validation study, which focused on wrist and forearm movements [14].

### 4.5. Further Research

The purpose of this study was to validate the LMC data specifically for finger kinematics as this device is mainly used for gesture-based applications. Our experimental protocol included only healthy subjects, and therefore, before issuing specific clinical guidelines, further research is required to replicate this study in different pathological populations and conditions. This study is part of a larger investigation, where, grasping tasks are assessed using the low-cost LMC with clinical feasibility as the focus. We are currently in the process of evaluating the performance of the LMC in a clinical environment among tetraplegic subjects.

Finally, the LMC development team is constantly improving and producing new versions of their software. These developments contain a more robust computer vision software and tracking algorithms paired with virtual reality and haptics. Future studies from our group will focus on larger data sets with a specific clinical application study design, which will give more detailed information with respect to the use of the LMC as a low-cost replacement for clinical evaluation tools.

## 5. Conclusions

The LMC device has the potential to replace maker-based motion capture technology; however, it requires significant programming knowledge to extract data from the device for offline analysis. Along with this paper, we provide the software for the data extraction, which minimises the need for significant programming knowledge. The major problem with the LMC is the variable sampling frequency, which is yet to be resolved. Methods such as Kalman filtering and dynamic time warping may be useful to circumvent this drawback. Finally, we suggest that the LMC provides agreeable data in static conditions; however, for dynamic cases, further validation studies should be conducted before it can be recommended as a viable alternative for marker-based motion capture in clinical environments.

## Figures and Tables

**Figure 1 sensors-21-01750-f001:**
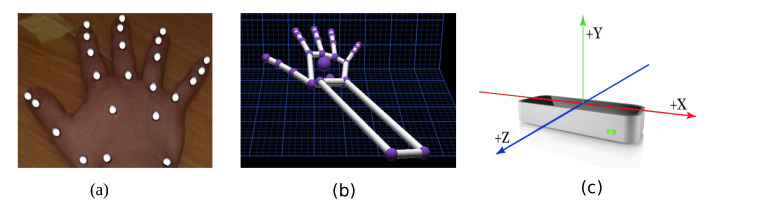
This figure shows the representation of the components used for the experiment. (**a**) Placement of the markers on the hand; (**b**) shows the skeletal hand model from the LMC; and (**c**) shows the LMC device with the axis orientation, which is the same for QTM.

**Figure 2 sensors-21-01750-f002:**
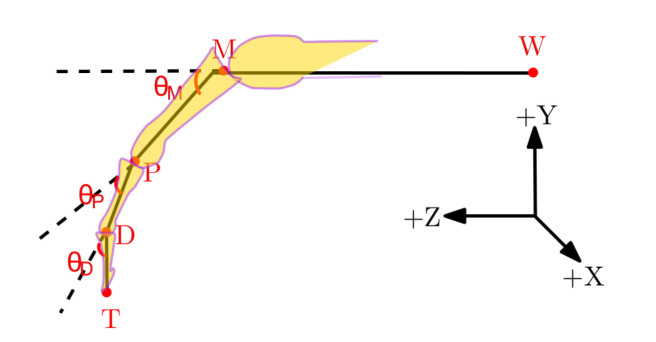
Geometrical sketch of a finger from the wrist joint depicting joint angles within the XYZ coordinate frame. The +X axis is from the ulna to radial styloid, the +Z axis from the midpoint of the wrist joint *W* to the metacarpal joint *M* and the +Y axis is the common line perpendicular to the X and Z axes.

**Figure 3 sensors-21-01750-f003:**
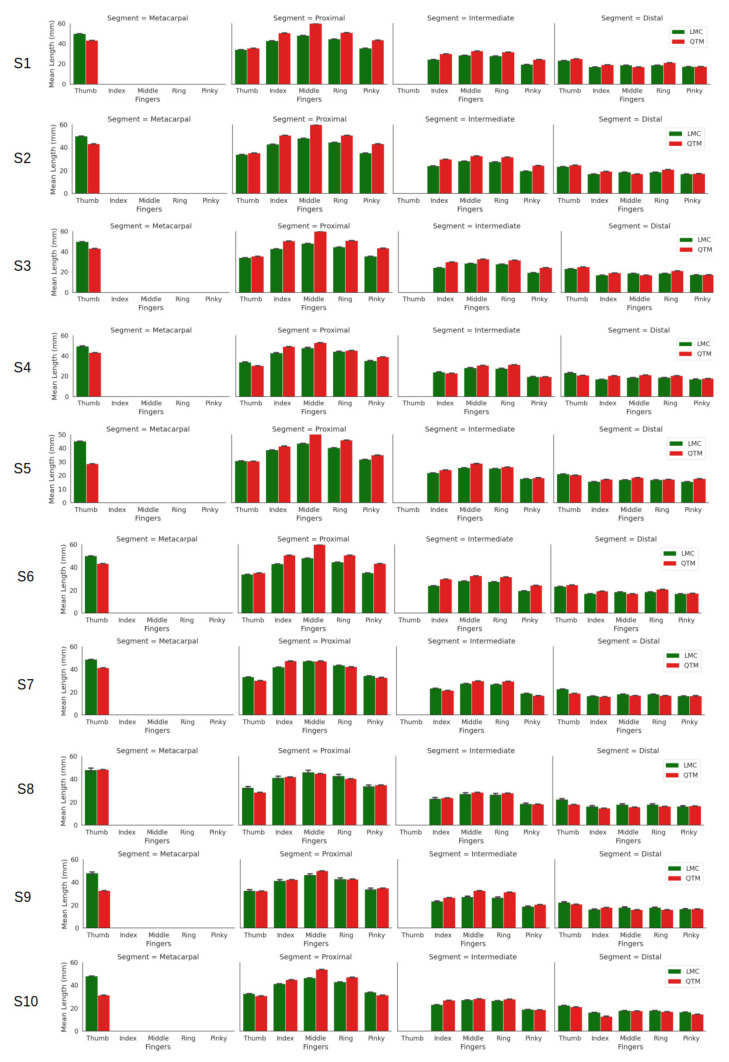
Segment lengths of the fingers of all ten subjects.

**Figure 4 sensors-21-01750-f004:**
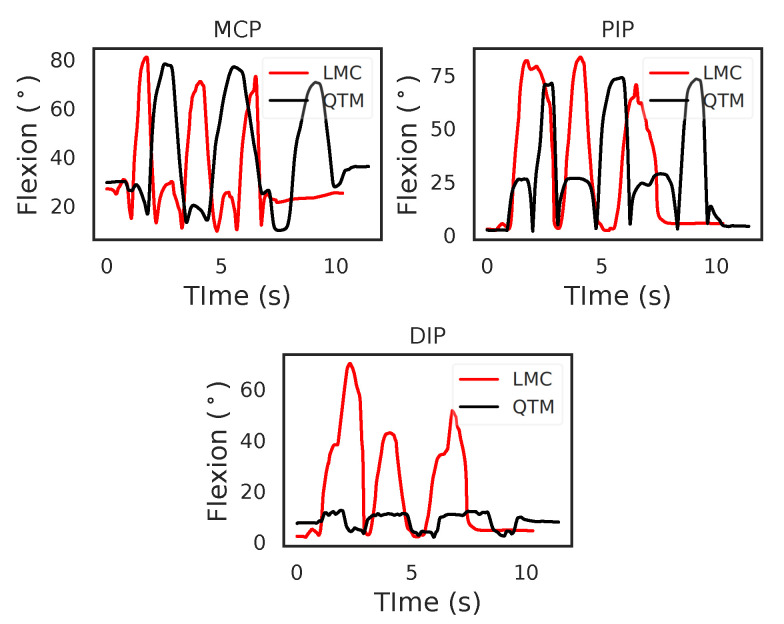
An example comparing the flexion trajectories of the LMC and QTM. This example is of index flexion motion, showing the flexion angles of the MCP, PIP and DIP joints.

**Figure 5 sensors-21-01750-f005:**
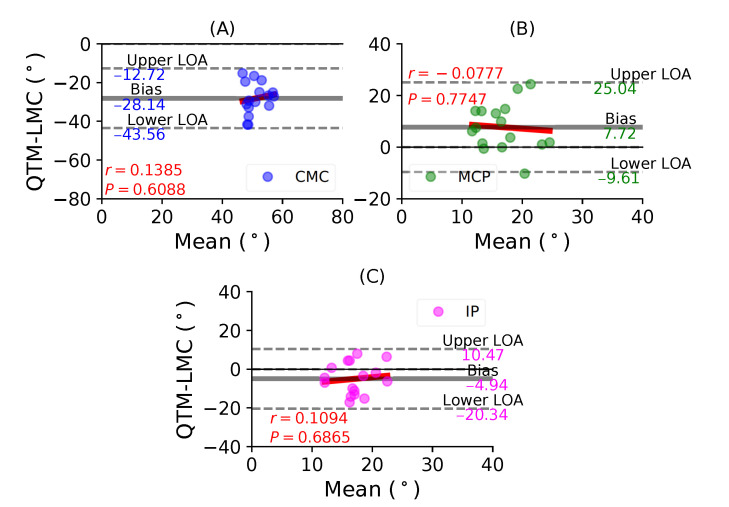
A BA plot of thumb abduction. (**A**) CMC joint, (**B**) MCP joint and (**C**) IP joint.

**Figure 6 sensors-21-01750-f006:**
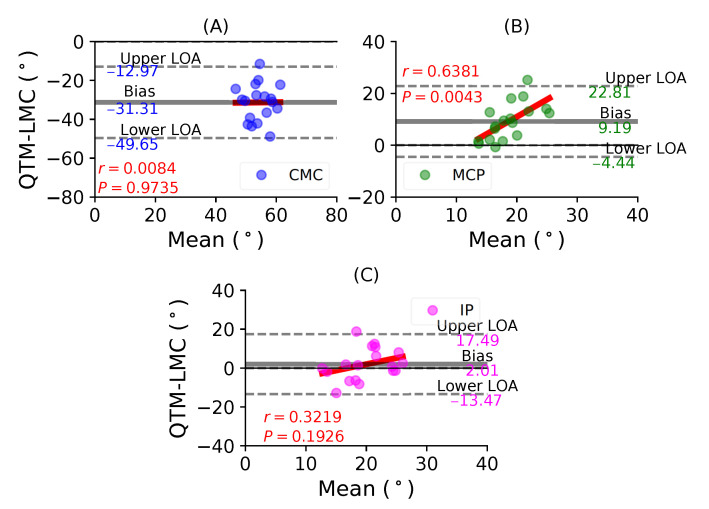
A BA plot of thumb flexion. (**A**) CMC joint, (**B**) MCP joint and (**C**) IP joint.

**Figure 7 sensors-21-01750-f007:**
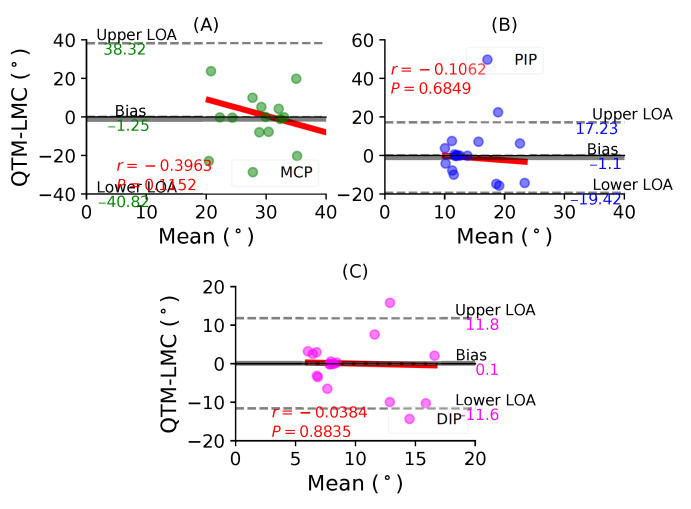
A BA plot of index finger flexion. (**A**) MCP joint, (**B**) PIP joint and (**C**) DIP joint.

**Figure 8 sensors-21-01750-f008:**
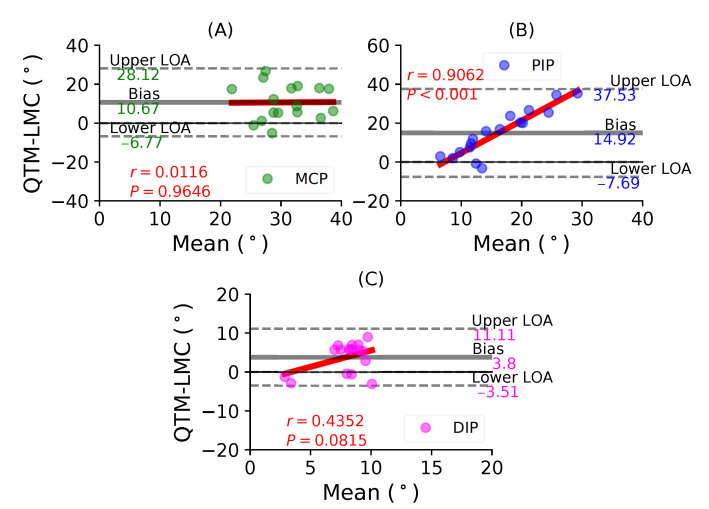
A BA plot of middle finger flexion. (**A**) MCP joint comparison, (**B**) PIP joint and (**C**) DIP joint.

**Figure 9 sensors-21-01750-f009:**
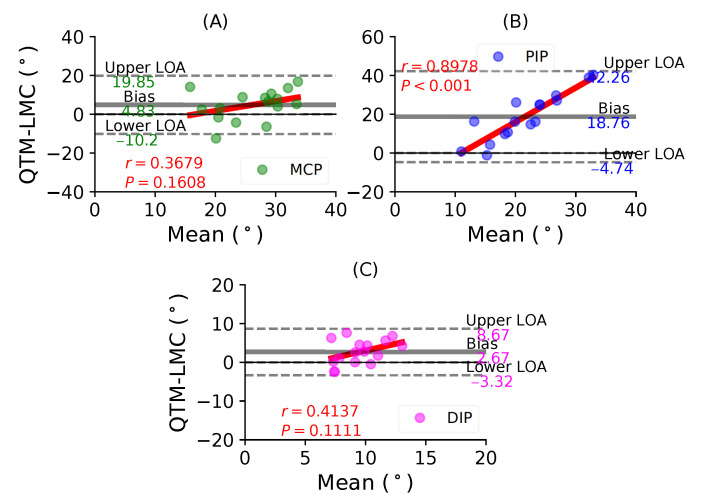
A BA plot of ring finger flexion. (**A**) MCP joint comparison, (**B**) PIP joint and (**C**) DIP joint.

**Figure 10 sensors-21-01750-f010:**
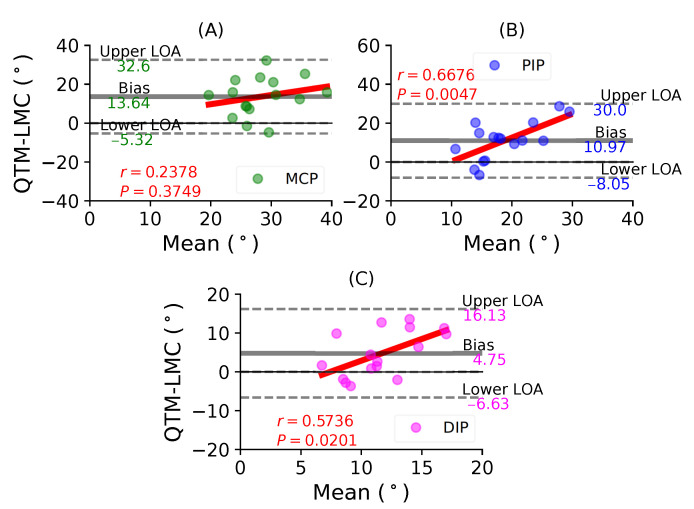
A BA plot of pinky finger flexion. (**A**) MCP joint comparison, (**B**) PIP joint and (**C**) DIP joint.

**Table 1 sensors-21-01750-t001:** Overall mean ± the s.d. of the segment lengths of ten subjects.

Fingers	Segment	LMC (mm)	QTM (mm)
Thumb	Distal	22.68 ± 10.61	19.27 ± 10.52
	Proximal	33.06 ± 10.70	32.32 ± 11.65
	Metacarpal	48.40 ± 12.41	38.12 ± 11.15
Index	Distal	18.99 ± 10.25	17.08 ± 9.72
	Intermediate	23.42 ± 10.69	25.73 ± 12.14
	Proximal	39.17 ± 12.86	44.89 ± 13.19
Middle	Distal	18.21 ± 12.08	17.71 ± 11.57
	Intermediate	27.58 ± 11.51	30.21 ± 12.07
	Proximal	46.46 ± 13.24	50.59 ± 13.07
Ring	Distal	18.10 ± 10.84	17.40 ± 10.75
	Intermediate	26.87 ± 10.56	28.95 ± 10.94
	Proximal	43.33 ± 11.59	44.90 ± 10.90
Pinky	Distal	16.67 ± 8.50	16.69 ± 8.65
	Intermediate	18.96 ± 8.41	19.58 ± 7.96
	Proximal	34.28 ± 2.76	36.31 ± 3.58

**Table 2 sensors-21-01750-t002:** Functional range of motion of the fingers at every joint for all trials.

Fingers	Joint	LMC (${}^{\circ }$)	QTM (${}^{\circ }$)	QTM-LMC (${}^{\circ }$)
Thumb Abduction	IP	31.26	37.75	6.49
	MCP	41.38	49.65	8.27
	CMC	54.10	42.44	–11.66
Thumb Flexion	IP	43.86	72.15	28.29
	MCP	43.16	62.47	19.31
	CMC	53.66	45.73	–7.93
Index	DIP	68.78	68.31	–0.47
	PIP	81.09	81.34	0.25
	MCP	107.86	86.66	–21.2
Middle	DIP	33.80	21.38	–12.42
	PIP	43.59	105.10	61.51
	MCP	72.36	92.64	20.28
Ring	DIP	47.49	19.86	–27.63
	PIP	69.55	103.94	34.39
	MCP	78.89	83.39	4.5
Pinky	DIP	41.16	50.32	9.16
	PIP	57.35	94.68	37.33
	MCP	78.08	95.61	17.53

## Data Availability

Synchronisation code and raw data are available online at this repository https://github.com/Amartya32/Leap_and_Motion_Capture (accessed on 9 November 2020).

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
