# Peer review of "Comparison of the Performance of the Leap Motion ControllerTM with a Standard Marker-Based Motion Capture System"

_sensors, 2021, doi:10.3390/s21051750_

Round 1

Reviewer 1 Report

The article offers a quantitative comparison between Leap Motion Controller (LMC) and Qualisys Track Manager (QTM). LMC is an inexpensive, model-based tracker of human hands which is a consumer electronic device and costs 2-3 orders of magnitude less than QTM which is a general tracker that is not model based, is industrial grade, and commonly used in medical applications and film industry. More importantly, LMC is a human-in-the-loop controller but QTM is a measurement/tracking device which makes this study slightly controversial. In other words, I am not completely convinced that the motivation is valid but assuming some researchers may misuse LMC as a measurement device, I am inclined to accept the paper with a few minor revisions.  

  1. Page 1, lines 19-21 refer to prior arts [1-6] citing the application of LMC in robotics, surgery etc. but misses LMC's application in hand rehabilitation and occupational therapy after stroke or other neurodegenerative disorders. Adding a few notable articles from that space may reach and benefit a wider audience. 
  2. Figure 2 shows the concept of interpolation; it is safe to assume that the readers are familiar with interpolation hence this figure can be removed. If not removed, it may be replaced with a zoomed-in version that shows how irregular the time stamps were in the LMC data and how new data points are interpolated at regular time stamps. Along the same line, Figure 3 is too large for the information it conveys.
  3. Lines 143 to 146 refer to Table 1 and state that the standard deviation was smaller in QTM data than it was in LMC data. A quick look at the table does not confirm that; in fact out of the 15 SDs reported in the table, there are several cases where the SD is smaller in the LMC data and a few are too close. A more accurate comparison and statement would help the readers.

The comparison of the movement data reported later in the article demonstrates a clear difference between LMC and QTM performance. This part of the research, in my opinion, matters significantly more to the research community.

Author Response

We thank the reviewer for their comments and suggestions which has made the manuscript better.

Reviewer 2 Report

The article presents the performance comparison between a very low-cost Leap Motion and a very expensive motion capture system. The conclusion of the paper is clear in that the Leap Motion cannot be recommended for clinical use due to low accuracy and repeatability. The reviewer thinks the conclusion of the authors is very natural as if we know the performance differences between a low-cost bicycle and a brand-new car. However, the authors have proven it in a wise way by using the statistical data obtained from the experiments. 

If you see figure 4, the segmental lengths of each finger show consistent results. The thumb length is estimated as a smaller value when the motion capture system is used, but the finger lengths are estimated as larger values. The reviewer thinks the skeletal kinematic model of Leap Motion is not correct compared with the model of the motion capture system. Indeed, it might not be related to the sensing capability of the Leap Motion. The kinematic modeling issue should be clarified prior to showing the resultant data.  

The reviewer cannot find the critical issues to be problems or questions to be resolved except that the ambiguity regarding the value as an academic paper. The followings are typos in the article 

1) In caption of figure 3: +Y axis => +Z axis 

2) In 176 line: Panel 6(B) => Panel 6(C)

3) In 271 line: are reflect => are reflecting 

4) In 275 line: The the 

Reviewer 3 Report

The authors compared the performance of the Leap Motion Controller with Standard Marker-Based Motion Capture System about their applications in healthcare services. The work is very important and results are logical to me, based on my limited knowledge. I thinks it is acceptable to publish in Sensors. The authors need to carefully consider the writing format of the figure legend, please check publications in Sensors and follow their writing format.

Reviewer 4 Report

The paper suffered from the poor performance of LMC.  The authors took risk to apply LMC to the finger tracking task without evaluating the accuracy of segment measurements and angular measurements of LMC, apart from quoting the reference 18 of "for static case, the standard deviation was less than 0.5 mm" and "the accuracy reduced by 5 mm with dynamic tracking". It would make sense that the authors could evaluate the accuracy and reproducibility of length and angular measurements of LMC in controlled conditions and check if LMC meet clinical acceptance before finger tracking experiments.

Fig 4 in fact did not show the errors of segment measurements clearly. It would be better to show the errors in a table, although this is not a fundamental issue. 

Round 2

Reviewer 2 Report

Although the conclusion of the paper was not interesting, the reviewer agrees that the authors have reflected all the comments in the revised version. Thank you for the effort.

Reviewer 4 Report

I accept the current version of the manuscript and have no further comments.